# Lactic Acid Bacteria as Antibacterial Agents to Extend the Shelf Life of Fresh and Minimally Processed Fruits and Vegetables: Quality and Safety Aspects

**DOI:** 10.3390/microorganisms8060952

**Published:** 2020-06-24

**Authors:** Sofia Agriopoulou, Eygenia Stamatelopoulou, Monika Sachadyn-Król, Theodoros Varzakas

**Affiliations:** 1Department of Food Science and Technology, University of the Peloponnese, Antikalamos, 24100 Kalamata, Greece; sagriopoulou@gmail.com (S.A.); estamatel@gmail.com (E.S.); 2Department of Chemistry, Faculty of Food Sciences and Biotechnology, University of Life Sciences in Lublin, 20950 Lublin, Poland; monika.sachadyn-krol@up.lublin.pl

**Keywords:** lactic acid bacteria, biopreservation, edible coatings and films, fresh-cut fruits and vegetables, shelf life extension, food quality, microbiological quality, postharvest, storage, bacteriocins

## Abstract

Eating fresh fruits and vegetables is, undoubtedly, a healthy habit that should be adopted by everyone (particularly due to the nutrients and functional properties of fruits and vegetables). However, at the same time, due to their production in the external environment, there is an increased risk of their being infected with various pathogenic microorganisms, some of which cause serious foodborne illnesses. In order to preserve and distribute safe, raw, and minimally processed fruits and vegetables, many strategies have been proposed, including bioprotection. The use of lactic acid bacteria in raw and minimally processed fruits and vegetables helps to better maintain their quality by extending their shelf life, causing a significant reduction and inhibition of the action of important foodborne pathogens. The antibacterial effect of lactic acid bacteria is attributed to its ability to produce antimicrobial compounds, including bacteriocins, with strong competitive action against many microorganisms. The use of bacteriocins, both separately and in combination with edible coatings, is considered a very promising approach for microbiological quality, and safety for postharvest storage of raw and minimally processed fruits and vegetables. Therefore, the purpose of the review is to discuss the biopreservation of fresh fruits and vegetables through the use of lactic acid bacteria as a green and safe technique.

## 1. Introduction

A healthy diet includes eating fruits and vegetables; their consumption is recommended by several government agencies because of their nutritional and medicinal properties [1] and low energy content [2]. Heart disease, colon cancer, obesity, and diabetes are some of the diseases that can be reduced with a high intake of fruits and vegetables [3]. Their consumption has increased in recent years, making them essential on a daily basis due to their abundance of nutrients [4]. In particular, in the last decade, the increased demand for fresh fruits and vegetables (whole and cut) in many industrialized nations has been covered partly from the production of minimally processed fruits and vegetables, as they are healthy and convenient foods. Traditional methods tend to be replaced by minimal processing methods, by providing fruits and vegetables that retain their quality for more than the usual time [5,6]. Two groups of molecules in the chemical composition of fruits and vegetables exist: nutritive molecules and non-nutritive phytochemicals [7]. Among nutritive molecules, vitamins, minerals, fibers, and micro and macronutrients are the most important, while phenolic compounds, flavonoids, and bioactive peptides belong to non-nutritive phytochemicals and have beneficial properties for human health [7]. The beneficial effect of all these molecules is proved in their action as receptors against free radicals [8]. According to the recent edition of Dietary Guidelines for Americans, published in 2016, from the U.S. Department of Health and Human Services (HHS) and U.S. Department of Agriculture (USDA), fruits and vegetables, should hold half the daily energy intake [9]. Moreover, vegetables with different colors (red, green, orange) from all categories, and whole fruits, are key recommendations in a healthy eating pattern. According to the International Fresh-Cut Produce Association (IFPA), fresh-cut produce is defined as “any fruit or vegetable or combination thereof that has been physically altered from its original form, but remains in a fresh state” [10]. Fresh-cut fruits and vegetables can be washed, trimmed, peeled, and chopped, creating 100% easy-to-use products that are still fresh, maintaining all of the characteristics during packaging (without further processing) and under refrigeration [11]. ‘Ready-to-eat’, ‘fresh-cut’, ‘easy- to-use’, or ‘pre-cut produce’ are some others designations used for minimally processed fruits and vegetables [12].

Fresh-cut fruits and vegetables are extremely perishable products that have a very short shelf life and physiological deterioration; biochemical changes and microbial degradation can occur during their marketing. All of these changes can cause significant degradation in quality of characteristics, such as color, aroma, and taste, and lead to the growth of undesired and harmful pathogens, limiting shelf life [5]. In addition, even minimal processing of fresh fruits and vegetables, such as cutting and peeling, leads to the leakage of cellular content around the injuring points, increasing the risk of a microbial infection, as these points are full of minerals, sugars, vitamins, and other nutrients [13,14]. Moreover, upcoming rapid tissue aging can significantly reduce the life of fresh-cut fruits and vegetables [15]. Browning, softening, and off-flavor development are some of the signs that may appear in fresh fresh-cut fruits and vegetables [16]. Mechanical wounding of fresh fresh-cut fruits and vegetables also increases the rate of respiration, which is directly connected with short postharvest life [17]. These products are ready for consumption without any further possible microbiological treatment, so quality and safety issues are very urgent for consumer health [6]. Damage to the outer surface of cut fruits and vegetables favors the survival and proliferation of foodborne pathogens, especially at temperatures above 4 °C. Lower temperatures can ensure a reduction in dynamic multiplication, but do not completely stop the survival of some microorganisms [18]. Microbial growth, in the case of fresh-cut fruits and vegetables, is significantly favored by the high water content of a large number of chopped tissues, as well as by the low or neutral pH that has the most vegetables and fruits, respectively [19].

In order to prolong storage life of fresh and minimally processed fruits and vegetables, many physical, chemical, and biological means and treatments have been proposed. Disinfection and washing are the main procedures used to reduce the population of pathogenic microorganisms (including their effect on the safety and quality of fresh and minimally processed fruits and vegetables). Among them, chlorine is a predominant treatment, which is added to water used to wash fresh-cut fruits and vegetables, although it has limited antimicrobial efficacy as it can only achieve 1–2 logarithmic reductions in pathogenic microorganisms [20]. The use of chlorine as a sanitizing agent poses serious risks to human health due to production of carcinogenic halogenated compounds [15]. European countries, such as Germany, Switzerland, the Netherlands, Denmark, and Belgium have taken into account all of the health issues that have arisen, and have banned the use of chlorine in disinfection of fresh and minimally processed fruits and vegetables. Another widely used disinfection practice is the use of NaClO with 50–150 mg L^−1^, which also has potential risks [21]. Different chemical alternatives to chlorine have been used, such as chlorine dioxide, and acidified sodium chlorite. Moreover, other substances have been used for the same purpose, such as ozone, organic acids, peroxyacetic acid, hydrogen peroxide, electrolyzed water, and calcium-based solutions [21,22,23,24,25,26].

Physical alternatives, such as ultraviolet light C, low-temperature storage, modification of atmosphere, and ultrasound or high pressure inert gas, to maintain quality and prolong shelf- life, have also been used [4]. As the cutting operations are unavoidable for fresh-cut fruits and vegetables, and the risk of microbial growth is possible, the avoidance of food-borne pathogen contamination is necessary without the production of potentially toxic substances. Nowadays, chemical compounds that are used for fresh-cut fruit and vegetable preservation are not preferable for consumers when they are eating fresh-cut fruits and vegetables, as they prefer healthier, more natural, safer, and non-chemically contaminated foods [6].

Food preservation with the help of natural antimicrobial agents could be a very promising technique, playing an important role in maintaining food quality and safety [27]. Food biopreservation is an alternative and novel method of preservation with increasing special interest from the consumers [22,28]. Biopreservation can extend the shelf life of fresh-cut fruits and vegetables by the use of safe, natural, or controlled microflora, and non-toxic biologically active compounds [29], enhancing their safety [30]. Biopreservation can help the production of fresh-cut fruits and vegetables, with increased safety, excellent nutrition, overall quality, and improved shelf life through the use of lactic acid bacteria (LAB).

In the present review, recent developments in novel shelf life extension technologies applied to fresh and minimally processed fruits and vegetables—in order to maintain or improve the quality and safety of them (with the use of LAB as a biopreservation method)—will be discussed.

## 2. Lactic Acid Bacteria

The use of LAB play a dominant role in the fermentation of both food and feed [31], with health and nutritional benefits, and a very long history and safe use after consumption of fermented foods and beverages [32]. Taste and texture are the main (quality) characteristics of fermented foods that are enhanced with the addition of LAB [33]. Dairy products, fermented fruits and vegetables, meat-based products, and fermented beverages are the main fermented foods that involve LAB [34,35,36]. LAB exists in environments such as water, soil, sewage, plants, as well as in humans and animals [32]. In general, environments rich in available carbohydrates are ideal for the growth of LAB. Cavities of humans and animals are also favorable places for their growth [34]. LAB can be isolated from many raw fruits and vegetables, and then used against natural microbial populations [37].

LAB belong to different taxonomic groups of Gram-positive bacteria, with a common characteristic that produces lactic acid as the main (or sole) product during fermentation of carbohydrates [38,39]. They have rod- or coccus-shaped cells [40], do not form spores, and are anaerobic or microaerophilic and acid-tolerant organisms [41]. They are naturally present in several food products, from which can be isolated [42].

LAB are generally regarded as safe (GRAS) microorganisms by the United States Food and Drug Administration (FDA), and Qualified Presumption of Safety (QPS) by the European Food Safety Authority (EFSA). Their use in food biopreservation are considered an alternative for the prevention of the growth of pathogenic microorganisms [43,44], as their competitiveness against pathogenic microorganisms make them extremely ideal candidates for the development of bioprotective agents for fresh fruits and vegetables [45]. During biopreservation, either antimicrobial metabolites can be applied without the producing strain, or culture-producing antimicrobial metabolites can be added [46]. These starter cultures can be added, either as individual cultures or as multi-species consortia [47]. In the group of LAB bacteria, there are 6 families, 38 genera, and all belong to the *Lactobacillales* order, *Bacilli* class, and *Firmicutes* phylum.

*Lactococcus*, *Streptococcus*, *Lactobacillus*, *Pediococcus*, *Leuconostoc*, *Lactosphaera*, *Melissococcus*, *Microbacterium Propionibacterium*, *Enterococcus*, *Carnobacterium*, *Tetragenococcus*, *Aerococcus*, *Alloiococcus Oenococcus*, *Vagococcus*, *Dolosigranulum*, and *Weisella* are the most common genera that belong to LAB [48,49,50,51]. Among all the genes present in LAB, Lactobacillus consists of 261 species (March 2020), ranking it in the genus with the most members. The genus *Lactobacillus* has been reclassified into 25 genera, including the *Lactobacillus delbrueckii* group, *Paralactobacillus*, and 23 novel genera with the names *Holzapfelia*, *Amylolactobacillus*, *Bombilactobacillus*, *Companilactobacillus*, *Lapidilactobacillus*, *Agrilactobacillus*, *Schleiferilactobacillus*, *Loigolactobacilus*, *Lacticaseibacillus*, *Latilactobacillus*, *Dellaglioa*, *Liquorilactobacillus*, *Ligilactobacillus*, *Lactiplantibacillus*, *Furfurilactobacillus*, *Paucilactobacillus*, *Limosilactobacillus*, *Fructilactobacillus*, *Acetilactobacillus*, *Apilactobacillus*, *Levilactobacillus*, *Secundilactobacillus* and *Lentilactobacillus* [52]. The following are the most common species: *Lactobacillus acidophilus*, *L. plantarum*, *L. Casei*, *L. rhamnosus*, *L. delbrueckii bulgaricus*, *L. fermentum*, *L. reuteri*, *Lactococcus lactis*, *Lactococcus lactis cremoris*, *Bifidobacterium bifidum*, *B. infantis*, *B. adolecentis*, *B. longum*, *B. breve*, *Enterococcus faecalis*, *Enterococcus faecium* [51].

LAB produce a variety of antimicrobial compounds, such as organic acids (lactic, citric, acetic, fumaric, and malic acid), hydrogen peroxide, CO_2_, diacetyl, ethanol, reuterin, acetaldehyde, acetoin, ammonia, bacteriocins, bacteriocin-like inhibitory substances (BLIS), and other important metabolites, which possess strong antagonistic activity against many microorganisms [53,54,55,56] (Figure 1). In addition, the antimicrobial effect of lactic acid bacteria is the result of competition with pathogenic microorganisms for nutrients [24].

Additionally, health-promoting properties have been linked with the presence of some strains of LAB and probiotics [57,58,59] as they have managed to reduce the risk of various diseases [60]. Probiotics have been identified as living microorganisms that have beneficial effects for humans and animals after adequate intake [61]. Probiotics have been used to prevent colon cancer [62], antibiotic-associated diarrhea, cholesterol reduction, lactose digestion [59], inflammatory bowel disease, breast cancer, and ulcerative colitis [63]. The genus *Lactobacillus* is one of the most widely used probiotics available on the market [60]. Probiotic bacteria do not live apart from the environment, but interact with the host, forming cooperative communities called biofilms [62].

In exception for their antimicrobial activity, LAB also have antifungal activity, which is of great interest, both against mycotoxigenic fungi and fungal mycotoxins, showing their potential by inactivation, removal, or detoxification processes [64,65,66]. The antifungal activity of LAB has prolonged the shelf life of fresh vegetables [67] and fruits [68].

## 3. Foodborne Pathogens on Fresh and Minimally Processed Fruits and Vegetables

Pathogenic microorganisms can infect fresh fruits and vegetables throughout the production process, starting from the soil, inadequately composted manure, contaminated seeds, contaminated water, and ending up to the foodstuff handlers, such as farmers, consumers, and kitchen workers—particularly the workers’ hands [69]. In Figure 2, all of the basic routes of contamination, of fresh produce by foodborne pathogens, are presented.

*Escherichia coli* O157:H7, *Salmonella spp*. [21], and *Listeria monocytogenes* are the main foodborne pathogens that exist on the surface of cut fruits and vegetables, causing serious damage to human health [18,70]. Outbreaks of these bacteria have been associated with fresh produce worldwide. In the U.S., most of these are linked with apples, stone fruits, mangos, blueberries, and papayas [71,72,73,74,75,76]. A recent outbreak was reported in 2019; it was associated with a *Salmonella* infection and linked to pre-cut melons [77]. Bacteria, such as *Aeromonas hydrophila*, *Bacillus cereus*, *Clostridium spp*., *Shigella spp*., *Vibrio cholerae*, *Campylobacter spp*., and *Yersinia enterocolitica,* have also been associated with high risk of illness outbreaks after people ate fresh and minimally processed fruits and vegetables, which has raised great concern [78]. In addition, parasites (*Cryptosporidium*, *Cyclospora*, helminths) and viruses (hepatitis A, noroviruses) are also a source of danger [79]. Table 1 presents some recent outbreaks associated with foodborne pathogens in fresh produce.

Salmonella bacteria are Gram-negative and can survive in a variety of conditions, with an optimum of 37 °C, a pH of 6.5–7.0, and a water activity of less than 0.94 [80,81], causing significant public health burdens in many countries. Salmonellosis is a foodborne disease, mainly caused by several non-typhoidal *Salmonella enterica* serovars, primarily the serovars *Enteritidis* and *Typhimurium* [82].

*Salmonella* spp. usually causes nausea, vomiting, abdominal pain, diarrhea, fever, and enterocolitis as main symptoms (after the consumption of contaminated foods) [83]. More than one million annual foodborne infections in the United States are caused by *Salmonella* spp., representing one third of all annual reported foodborne bacterial infections [84]. Reports show that *Salmonella* spp. is the most common bacterial pathogen responsible for fresh produce-associated disease outbreaks in developed countries, accounting for at least half of these outbreaks in the European Union (50%) [85].

The biofilms formed by many strains of the *Salmonella* spp. on various processing-related surfaces, such as glasses, plastics, wood, and metals, may cause the so-called cross-contamination in different vegetables [86]. The biofilm functions as a survival mechanism as it helps microorganisms propagate in the environment, and to resist antimicrobial and sanitizing agents. Polysaccharides, proteins, lipids, and extracellular DNA are the main components of the biofilm. In exception, food biofilms can be found on moist surfaces, water pipelines, and pathological human tissues and organs [87].

According to an EFSA summary report, in 2016, salmonellosis was classified as a foodborne infection, with the highest incidences in the European Union (EU) when compared to other foodborne illnesses [88].
microorganisms-08-00952-t001_Table 1Table 1Recent outbreaks associated with foodborne pathogens in fresh produce.Food MatrixPathogensReported CasesHospitalizationsRegionDeathsRecallYearReferenceCut fruits*Salmonella Javiana*1657314 states of USA-Yes2020[89]Pre-cut melons*Salmonella Carrau*137 389 states of USA-Yes2019[77]Fresh papayas *Salmonella Uganda*81279 states of USA-No2019[90]Cucumbers*Salmonella Poona*90720440 states of USA6Yes2016[91]Mushrooms*Listeria monocytogenes*363017 states of USA 4Yes2020[92]Cantaloupes*Listeria monocytogenes*14714328 states of USA33Yes2012[93]Romaine lettuces *Escherichia coli* O157:H71678527 states of USA-Yes2020[94]Romaine lettuces*Escherichia coli* O157:H72109636 states of USA5No2018[95]Leafy greens*Escherichia coli* O157:H725915 states of USA1No2018[96]

The detection of *Salmonella spp*. in fresh-cut cabbage, sprouts, melons, cucumber, lettuces, tomatoes, carrots, and other raw fruits and vegetables, raises concerns about safety of consumption and consumer health. In addition, *Salmonella* Enteritidis was shown to be able to survive at 5 °C and grow at 10 and 20 °C in Red Delicious apple flesh [70].

*E. coli* is Gram-negative and causes intestinal infections and extra-intestinal illnesses, both in humans and animals, having diarrhea as a basic symptom [71,97]. Sprouts, lettuce, and spinach are the most associated vegetables with contamination from *E. coli*. The strain *E. coli* O157: H7 was the first reported as enterohemorrhagic, having the ability of production of Shiga-like toxins [98]. *E. coli* O157: H7 can survive and grow, both in fresh fruits [70] and in fresh leafy green vegetables [98]. In the U.S., the most serious outbreak of *E. coli* O157: H7 associated with vegetables was recorded in 2018, when 5 of 210 patients died after consuming romaine lettuce [95]. In the European Union, one of the largest reported outbreaks of *E. coli* O104: H4 associated with sprouted seeds occurred in Germany and France (in 2011), with 3831 cases and 54 deaths [99].

*Listeria* are Gram-positive anaerobic bacteria that are ubiquitously found in nature. *L. monocytogenes* is the only bacterium that can cause human listeriosis through food consumption [43]. A century has passed since the 1920s, when listeriosis of humans and animals was first described as an infection [100]. As its occurrence in vegetables can reach 25%, it causes concern in human health due to the fact that these products are consumed raw, with the proven growth of *L. monocytogenes* under refrigerated and ambient conditions [43]. Severe symptoms and high fatality rates may occur in people with weak immune systems, neonates, and pregnant women [101,102]. According to the EFSA, in 2019 it was reported that *L. monocytogenes* caused 229 deaths out of 2549 confirmed cases in 2018. The category “vegetable and juices and other products thereof” was the food vehicle category with the higher percentage of listeriosis outbreak (28.6%), indicating that the category of food is an important source of human infection [103]. *L. monocytogenes* can survive in extreme conditions, such as in cold, humid, and low oxygen environments, and it has been isolated from a variety of raw fruits and vegetables [70,104]. Apples and stone fruits are some of the fruits with high incidences of *L. monocytogenes* [105].

## 4. Antimicrobial Effects of Lactic Acid Bacteria in Fresh and Minimally Processed Fruits and Vegetables

In recent years, there has been a trend towards using biological methods to reduce the contamination of fresh fruits and vegetables to pathogens. Therefore, bioprotective agents have been used to inhibit the growth of pathogens [106]. Except from antimicrobial compounds, the microbial polysaccharides produced by LAB also have shown significant antibacterial activity against pathogens, such as *E. coli*, *L. monocytogenes*, *Salmonella* Typhimurium and *Shigella sonnei* [107].

LAB belong to the most promising biocontrol agents for improving shelf life of fresh-cut fruits and vegetables. Although the shelf life of fresh fruits and vegetables can last a few weeks, this cannot happen with fresh-cut fruits and vegetables that are only kept for 4 to 10 days, under refrigerated conditions, due to the possible growth of pathogenic microorganisms [24].

The use of LAB in the biopreservation of fresh fruits and vegetables can also be enhanced when combined with other storage methods. Low oxygen modified atmosphere packaging is one of the well reported methods that is applied in storage for fresh produce for extending shelf life [108]. The combination of *Lactobacillus plantarum* subsp. *plantarum* CICC 6257 with low oxygen modified atmosphere packaging technology has been successfully used for the reduction of pathogenic potential of *L. monocytogenes* in cabbages, according to a recent study by Dong et al. [109].

Vescovo et al. [110] and Torriani et al. [111] first used the *Lactobacillus casei*, or their culture filtrate to control pathogenic microorganisms in ready-to-use vegetables, and highlighted their inhibitory effects. The addition of 3% culture permeate of *L. casei* IMPC LC34 to mixed salads was, according to Torriani et al. [111], reported to cause a decrease in the total mesophilic bacteria counts from 6 to 1 log colony forming units (CFU)/g and to suppress coliforms, enterococci, and *A. hydrophila* after 6 days of storage at 8 °C. In another study, the inhibitory activity of some lactic acid bacteria was evaluated against foodborne human pathogens. Strains of *Leuconostoc* spp., *Lactobacillus plantarum*, *Weissella* spp., and *Lactococcus lactis* were isolated from fresh fruits and vegetables and they were inoculated on wounded Golden Delicious apples and Iceberg lettuce cut leaves. The results showed the total inhibition of *L. monocytogenes* and the reduction in population of *S.* Typhimurium and *E. coli* by 1 to 2 log CFU/wound or g [19].

The enrichment of minimally processed yellow melon by *Lactobacillus rhamnosus* HN001 and the correlation between microbiological characteristics was evaluated by Martins de Oliveira et al. [13]. The results from this study demonstrated high viability of *Lactobacillus rhamnosus* HN001, as there were high adhesion of *L. rhamnosus* HN001 on the vegetal tissue; microbiological safety as *Salmonella* sp. was not detected in any sample, giving a healthy, transformed probiotic food. In a more recent study, bacterium suspension belonging to *L. plantarum* was sprayed in fresh lotus roots in order to limit the oxidation of phenolic compounds that are responsible for enzymatic browning reaction, and to evaluate the postharvest properties of this vegetable. The *L. plantarum* suspension proved capable of causing an 84.17% transformation of catechin after interaction for 30 h with plant skin. Texture characteristics, such as hardness, chewiness value, springiness, and cohesiveness, also significantly improved, proving that the use of lactic acid bacteria have contributed to the extension of shelf life of fresh lotus roots [112].

Lokerse et al. studied the development of *L. monocytogenes* in the ingredients of fresh-cut salads by determining the effect of product characteristics and the presence of competitive flora, especially LAB. Most of the products tested did not have the presence of *L. monocytogenes* greater than 3.4 log CFU/g, and only the Galia melon exceeded 3.4 log CFU/g, indicating that the Galia melon, which is often used in fruit salads, is the main ingredient that can contribute to the development of *L. monocytogenes*. Inhibition of the growth of *L. monocytogenes* was achieved in some components of fruit salads, such as non-pasteurized potatoes, white cabbage, and mango containing a high number of LAB [101].

Recently, Ramos et al. (2020) developed an alternative biopreservation approach to maintain the safety of fresh lettuce, rocket salad, parsley, and spinach. As a protective culture against *L. monocytogenes*, they examined the potential of bacteriocinogenic LAB, *Pediococcus pentosaceus* DT016 during preservation. In vegetables inoculated with *P. pentosaceus* DT016, the number of pathogens were significantly lower (*p* < 0.01), and only at the last day of storage, a minimal difference of 1.4 log CFU/ g was observed compared to vegetables without protective culture [43]. Antagonistic effect of *Pseudomonas graminis* CPA-7 against two foodborne pathogens (*Salmonella* spp. and *L. monocytogenes*) was recorded in fresh-cut apples [113] and melons [114].

In Table 2, studies with LAB and biopreservation of fresh and minimally processed fruits and vegetables are presented.

## 5. Antimicrobial Effects of Metabolites of Lactic Acid Bacteria in Fresh and Minimally Processed Fruit and Vegetables

### 5.1. Bacteriocins

Bacteria (Gram-positive and Gram-negative) produce several interesting substances of protein structure, with bactericidal or bacteriostatic action, called bacteriocins, which act competitively for the same ecological position, or nutrient pool, with closely related microorganisms, and mainly with Gram-positive bacteria [120,121,122]. These substances are used by bacteria as a defense against other threatening microorganisms [123]. Anti-viral and anti-fungal properties can also be caused by some bacteriocins [124]. Bacteriocins are different from antibiotics, as they are produced during the lag phase and are classified as primary metabolites, while antibiotics are produced after the end of microbial growth and are classified as secondary metabolites [125]. There are two different types of bacteriocins, according to their inhibitory spectrum. In the first type, bacteriocins exert their inhibitory activity against bacteria of the same species with the bacteriocin producing bacterium; they are called narrow spectrum bacteriocins. In the second type, bacteriocins exert their inhibitory activity against bacteria of different genera; they are called broad-spectrum bacteriocins [124]. A bacteriocin is considered ideal when small concentrations are required to act, has a wide range of action against several pathogenic microorganisms, does not cause any damage to the product that is applied, and there is no high cost for its production [126].

Bacteriocins produced by LAB, are antimicrobial peptides, containing about 30–60 amino acids, with small molecular weight that can be used as natural and safe food preservatives in a variety of foods, including fruits and vegetables, without altering the nutritional and sensorial properties or the physicochemical characteristics of the food [49,104]. There are also bacteriocins that are larger in size and are described as proteins [127]. They are ribosomally synthesized peptides; they have different structure and biochemical properties. Moreover, they are differentiated both in the mode of action and spectrum of activity [128,129], in size, and in specificity [126]. Preservation with bacteriocins belong to the non-thermal preservation method [130]. There are many bacteriocins, such as nisin, enterocin AS-48 and 416K1, bovicin HC5, enterocin 416K1, bificin C6165, and pediocin, which have been described and tested in the preservation of quality in fresh fruits and vegetables [28,129]. The mechanism by which antimicrobial activity is performed is relatively complicated, and for each bacteriocin and food, different types of interactions are created [131]. Bacteriocins act against the bacterial cytoplasmic membrane by disrupting the movement of protons [49], causing the leakage of ions, ATP, RNA or DNA molecules and the cellular death of spoiling and pathogenic microorganisms [132].

Bacteriocins have poor efficiency towards Gram-negative bacteria, as this group of bacteria have an outer membrane, which acts as an efficient permeability barrier [133]. This membrane prevents the passage of molecules, including antimicrobials agents [134]. Moreover, the efficacy of bacteriocins against Gram-negative bacteria can be enhanced by the addition of chelating agents (e.g., EDTA) or hydrostatic pressure [120,135]. Bacteriocins are active in the nanomolar range [128]; they have high-temperature stability [136] and, until 2017, had quantified a total of 785 bacteriocins from LAB [38]. Various factors affect the production of bacteriocins, including environmental factors, temperature, and pH. Temperature 30–37 °C and pH 5.0–8.0 are the optimum conditions in which bacteriocins are usually developed [123,128]. The reduced effectiveness of bacteriocins can be attributed to the development of resistance by certain pathogenic microorganisms, to their interaction, inactivation, or even to their binding to a variety of food ingredients, as well as to their random distribution in the food [137]. Bacteriocins act against a limited number of target-bacteria (which usually have the same needs) and this is the main disadvantage of these antimicrobial substances [138].

From the first classification of bacteriocins in 1993 by Klaenhammer [139], many classifications have been mediated. One of the last classifications has been made by Alvarez-Sieiro et al. in 2016. This classification was based on the biosynthesis mechanism, genetics, and structure; according to it, bacteriocins were classified into three classes, heat stable (<10 kDa) Class I, and Class II, and thermolabile (>10 kDa) Class III [38]. Class I bacteriocins, Class II bacteriocins, and Class III bacteriocins are also called lantibiotics [134], non-lantibiotics, or pediocin-like antibiotics, and are sensitive to heat, respectively [140,141].

The use of bacteriocins as an alternative method of preserving fruits and vegetables is under investigation, although the bacteriocins have the properties to be used for their preservation. To date, only nisin (marketed as Nisaplin and other brand names) and pediocin PA-1/AcH and Micocin® have been approved for use as food additives by the FDA. Although their use has not been formally approved in fruits and vegetables, many studies have evaluated their use in fruits and vegetables as they act as natural antimicrobials and alternatives to chemical food preservatives [129,142]. In fact, an indirect way of introducing bacteriocins into foods is being implemented, so that producer strains can be inoculated to fresh produce, to produce bacteriocins in situ [38]. Moreover, bacteriocins can be added directly to fruits and vegetables, acting as food additives, which increases their microbiological safety [143].

Dhundale et al. (2018) examined the use of bacteriocinogenic lactic acid bacteria (BLAB) for biopreservation of fruits. Bacterial-producing LABs were isolated from curd and cow dung samples and were tested on 25 isolated bacterial fruit flora from apple, fig, banana, sapodilla, kiwi, strawberry, and pomegranate. The BLAB coating forms a film on the surface of the fruit; thus, inhibiting the bacteria that destroy the fruit. The antibacterial activity against the fruit flora, as well as the pathogenic microorganisms *E. coli, Staphylococcus aureus, Bacillus cereus, Pseudomonas aeroginosa, Proteus vulgaris, Salmonella typhi, Serritia* spp, *Xanthomonas campestris,* make the use of BLAB a bioproduct capable of prolonging fruit life [144]. Table 3 presents studies on the biopreservation of fresh and minimally processed fruits and vegetables with the use of bacteriocins.

### 5.2. Nisin

Nisin is the most studied and applied bacteriocin and has been characterized as GRAS by the Food and Agriculture Organization (FAO) of the United Nations, the World Health Organization (WHO), and the FDA. In addition, the use of nisin in certain foods is authorized as a preservative additive (E-234) in the European Union (EU) [152]. Nissin was the first commercially used bacteriocin since 1969. The lethal dose (LD_50_) has been estimated as 6950 mg/kg, which means that there are no toxic effects on humans [153]. In context of the GRAS status, the maximum allowable dose of nisin that can be used is 250 ppm [154]. The combined action of nisin with various organic acids could contribute to increasing its inhibitory effect against various Gram-positive bacteria that have developed resistance to this bacteriocin [155]. Nowadays, its use as a bioprotective agent in the food industry has been allowed in more than 50 countries worldwide [156]. Nisin is effective, mainly against Gram-positive foodborne pathogens or spoilage bacteria, such as *S. aureus*, *L. monocytogenes*, *A. acidoterrestris*, *Clostridium,* and *Bacillus* spores [129,157]. Except for the antibacterial ability of nisin, it has been proven that nisin has positive effects in the maintenance of vitamin C, in minimally processed mangoes, meaning that nisin can help in the quality of this fruit [158].

Nisin is a hydrophobic and cationic protein. It has low molecular weight (3353 Da), and is found in several types, such as, nisin A, nisin U, nisin Z, etc. Every type has different antimicrobial activity spectrum, but all are produced by certain strains of *Lactococcus lactis* ssp. *lactis* [123].

The combination of nisin-producing *Lactococcus lactis* CBM21 with thyme essential oil showed good inhibitory effect against *L. monocytogenes* and *E. coli* when applied as washing solution in the lamb’s lettuce without affecting their quality parameters [159].

### 5.3. Pediocins

Pediocins are produced mainly by strains of *Pediococcus pentosaceus* and *Pediococcus acidilactici* from the *Pediococcus* spp. genus [160]. Pediocins are basically characterized by their antilisterial activity [161,162]. Pediocin can be used in antimicrobial packaging, as it can be incorporated with films. Nanocomposite films with pediocin and ZnO nanoparticles presented antimicrobial activity against *S. aureus* and *L. monocytogenes* [163]. A pediocin DT016 solution was used to inactivate *L. monocytogenes* in fresh, leafy vegetables (lettuce, rocket salad, parsley, and spinach) during prolonged storage at 4 °C. The results were compared with those of leafy vegetables washed with water and chlorine. In pediocin-washed vegetables, there was a significant reduction of pathogen proliferation, by 3.2 and 2.7 log CFU/g, compared with vegetables washed in water and chlorine, respectively [43].

## 6. Edible Coatings and Films, and Lactic Acid Bacteria in Fresh and Minimally Processed Fruits and Vegetables

The use of edible coatings and films in fresh-cut fruits and vegetables aims at maintaining the overall quality, as well as extending shelf life, as several active ingredients can be incorporated into these innovative and alternative materials, controlling basic functions, such as moisture transfer, gas exchange, and oxidation processes. Texture enhancers, antioxidants, nutrients, antimicrobial, and anti-browning agents can be added in fresh-cut fruits and vegetables, without affecting consumer acceptance, contributing significantly to quality improvement and safety [164].

The surface of minimally proceeded fruits and vegetables is undoubtedly the most contaminated part with foodborne pathogens. The use of bacteriocin-incorporated edible coatings and films could affect the quality and safety of these products, reducing the disadvantage of interacting with ingredients that fruits and vegetables have [129]. The incorporation of live microbial cells of LAB with edible coatings and films can affect some physical properties, such as barrier and mechanical, and can also bind the antimicrobial substances to the food product in order to maintain a proper cell concentration that will be able to exhibit an antimicrobial effect [165].

Edible coatings and films are an alternative approach to maintaining quality and extending the shelf life of fresh-cut fruits and vegetables, and partially prevent the colonization of the fruit by human pathogenic bacteria, leading to increased food safety [166,167]. Edible coatings and films added with bacteriocins are promising natural preservatives for reducing foodborne pathogens [129], and the way of application is very important. Between direct application of bacteriocin and its inclusion in an edible coating and film, the second option appears the most promising, as the release of antimicrobial agent is controlled and the stability of the antimicrobial agent is also increased [158].

In addition, the combination of bacteriocins with other antimicrobial substances or stressors have managed to increase their antimicrobial efficacy against spoiling and pathogenic microorganisms [156]. Edible antimicrobial coatings are consumed with the fruit or vegetable that they wrap, while at the same time, being natural barriers to gas exchange between fresh produce and the environment [168]. In addition, edible coatings and films can incorporate with bifidobacteria in order to increase the probiotics properties of fresh produce [169].

Tenea et al. (2020), in a very recent study, investigated the efficacy of peptide-based coatings from *Lactobacillus plantarum* UTNCys5-4 and *Lactococcus lactis* subsp. lactis Gt28 strains against a pathogenic cocktail containing *E. coli*, *Salmonella* and *Shigella* in fresh-cut slices of pineapple. The results, after 5 days preserved in refrigeration, showed a decrease in cell counts of 2.08 log CFU/g, 1.43 log CFU/g, and 1.91 log CFU/g for the *E. coli*, *Salmonella,* and *Shigella,* respectively, indicating that these coatings are a good alternative to chemical compounds, increasing the shelf life and safety of fresh-cut pineapple [170]. Moreover, coatings with antimicrobial peptides from *Lactobacillus plantarum* UTNGt2 were used for inhibition of *E. coli* and *Salmonella* and the estimation of postharvest quality in fresh tomatoes. After 17 days of storage in room temperature, in the peptide-treated tomatoes, no external alteration was observed, and although no complete reduction of *E. coli* and *Salmonella* was observed, effective inhibition was demonstrated [171].

The enrichment of 0.5% carboxymethyl cellulose (CMC) edible coatings with bacteriocin from *Bacillus methylotrophicus* BM47 was studied by Tumbarski et al. (2019) in order to evaluate postharvest quality parameters and their efficacy of extending shelf life in fresh strawberries. Strawberries were preserved at 4 °C and relative humidity 75% for 16 days. The results were compared to CMC coated strawberries. The use of CMC with bacteriocin (CMC + B) coatings showed positive influence on antioxidant activity, improvement in commercial appearance, and in shelf life [28]. Table 4 presents studies on the biopreservation of fresh and minimally processed fruits and vegetables with the use of edible coatings and films.

## 7. Conclusions

Foodborne pathogens are inevitable microorganisms found in fresh-cut fruits and vegetables, mainly associated with foodborne outbreaks, such as listeriosis and salmonellosis. Although chemical control of these pathogens is widely used in the industry, consumer demand for healthy, “more nature”, and ‘free chemical” fresh produce has led the food industry to adopt biological control as an alternative technique to maintain safety and reduce contamination. Protective cultures of LAB and their metabolites play an important role in biocontrol, as their potential inhibitory effects against pathogens are well documented, without changing the sensory properties of foods. In particular, the use of primary metabolites, bacteriocins, both in situ after inoculation to the fresh produce, and by incorporation into edible coatings, is a simple and environmentally friendly biopreservation technique that does not require the use of expensive laboratory equipment. Moreover, the use of bacteriocins and edible films and coatings does not affect the organoleptic characteristics of fruits and vegetables and consumer acceptance. Additionally, as LAB can survive under cold storage temperatures, their metabolites can be used as food bioadditives in preservation of fresh-cut fruits and vegetables. Characteristics of the producer strain culture, such as technical effectiveness, commercial viability, the applied dose, and the complex mechanisms of action (including membrane permeabilization) are issues that need to be carefully addressed. As foods are complex matrices with varied components, the effectiveness of each biopreservative agent should be evaluated separately.

## Figures and Tables

**Figure 1 microorganisms-08-00952-f001:**
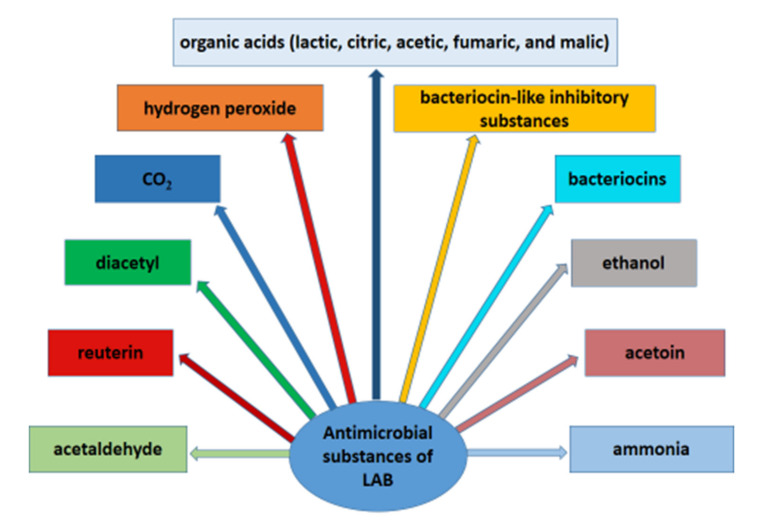
Antimicrobial substances produced by lactic acid bacteria.

**Figure 2 microorganisms-08-00952-f002:**
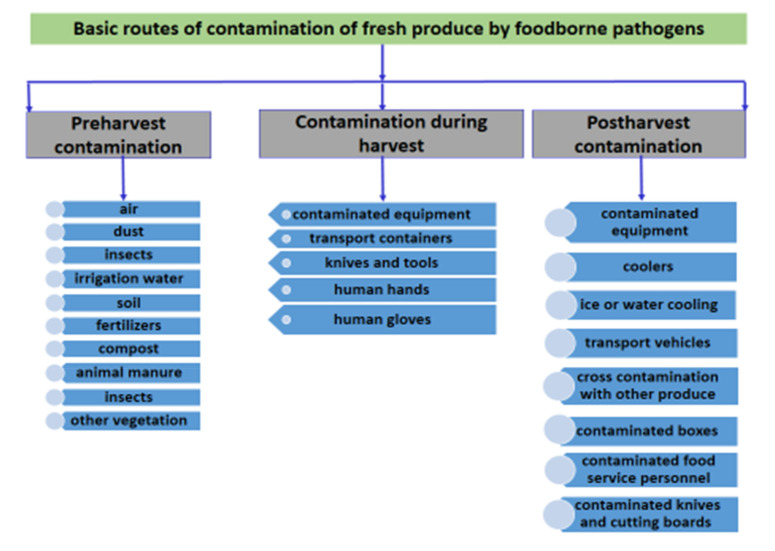
Basic routes of contamination of fresh produce by foodborne pathogens.

**Table 2 microorganisms-08-00952-t002:** Lactic acid bacteria and biopreservation of fresh and minimally processed fruits and vegetables.

Food Matrix	Lactic Acid Bacteria	Target Pathogen or Postharvest Properties	Process Duration	Effect	Reference
Fresh-cut pear	*Lactobacillus rhamnosus* GG	*Salmonella spp*. and *L. monocytogenes*	9 days	Reduction on *Salmonella spp*. population, no effect in *L. monocytogenes*	[115]
Lamb’s lettuce	*Lactobacillus plantarum, Lactobacillus casei*	*E. coli* and *L. monocytogenes*	16 days	Significant inhibition of *E. coli* and *L. monocytogenes*	[20]
Sliced apples	*Lactobacillus plantarum*				
Table grapes	*Lactobacillus delbrueckii* subsp. *bulgaricus* strain F17	Aerobic mesophilic bacteria, yeast and molds, and coliform bacteria ^a^	20 days	Significant inhibition during the storage period and improvement in the postharvest quality	[116]
	*Leuconostoc lactis* strain H52				
Fresh-cut curly leafy greens	*Lactobacillus curvatus*	*L. monocytogenes*	8 days	Reduction of *L. monocytogenes*	[102]
Fresh-cut fruit mixture	*Lactobacillus pentosus* MS031	*L. monocytogenes*, *E. coli*, *S. aureus*	10 days	Reduction 96.3 % of *L. monocytogenes,* undetectable level for *E. coli, S. aureus*	[117]
Fresh-cut cantaloupe	*Lactobacillus plantarum* B2, *Lactobacillus fermentum* PBCC11.5	*L. monocytogenes*	11 days	Reduction of *L. monocytogenes*	[118]
Lotus root	*Lactobacillus plantarum* (LH-B02)	Postharvest properties	15 days	Reduction of color loss enhancement of elasticity, coherence	[112]
Litchi	*Lactobacillus plantarum*	Postharvest properties	21 days	Reduction of browning, reduction of color loss, high concentration of phenolic compounds	[119]
Mixed salads	*Lactobacillus casei*	Coliforms, enterococci, and *Aeromonas hydrophila*	6 days	Reduction in the total number of mesophilic bacteria, suppression of coliform bacteria, enterococci and *Aeromonas hydrophila*	[110]

^a^ Not determined in the Publication.

**Table 3 microorganisms-08-00952-t003:** Bacteriocins and biopreservation of fresh and minimally processed fruits and vegetables.

Food Matrix	Bacteriocins	Target Pathogen	Process Duration	Effect	Reference
Fresh-cut leafy greens	Pediocin DT016	*L. monocytogenes*	15 days	Significant inhibition of *L. monocytogenes*	[43]
Fresh-cut lettuce	*Pseudomonas graminis* CPA-7 and nisin	*L. monocytogenes*	6 days	Reduction of *L. monocytogenes*	[106]
Fresh-cut lettuce	Bacteriocin	*L. monocytogenes*	6 days	Reduction of *L. monocytogenes*	[30]
Cabbage	Crude bacteriocin extracts from the *Lactobacillus* species	*S. aureus*, *E. coli,* and *Shigella* species	3 days	Inhibitory activity against *S. aureus, E. coli,* and *Shigella* species	[142]
Fresh-cut iceberg lettuce	Nisin A	*L. monocytogenes*	7 days	100-fold reduction of *L. monocytogenes*, extend the shelf life	[145]
Fresh strawberries, tomatoes and mushrooms	Bacteriocin, producing by *Pediococcus spp.*	*E. coli* and *Shigella spp.*	15 days	Increased shelf life and enhanced microbiological quality	[146]
Potatoes	Nisin-formic acid combination	*Bacillus subtilis*	10 days	Inactivation of the proliferation of *Bacillus subtilis*	[147]
Bananas	Enterocin KT2W2G-cinnamon oil combination	*Klebsiella variicola, Serratia marcescens, Lactococcus lactis subsp. Lactis, Klebsiella pneumoniae Enterococcus faecalis*	a	Inhibition spoilage bacteria and extension the shelf life of bananas	[148]
Fresh-cut melon	Nisin	*E.coli* O157:H7, *L. monocytogenes*	7 days	Reduction *E.coli* O157:H7 and *L. monocytogenes*	[149]
Fresh-cut lettuce	Nisin, coagulin and a cocktail of both bacteriocins	*L. monocytogenes*	7 days	Decrease in the viability of *L. monocytogenes*	[150]
Fresh fruits	Enterocin AS-48	*L. monocytogenes*	7 days	Significant inhibition or completely inactivation of *L. monocytogenes*	[151]

^a^ Not determined in the Publication.

**Table 4 microorganisms-08-00952-t004:** Edible coatings and films and biopreservation of fresh and minimally processed fruits and vegetables.

Food Matrix	Edible Coatings and Films	TARGET Pathogen	Process Duration	Effect ^a^	Reference
Fresh strawberries	*Bacillus methylotrophicus* BM47 incorporated into carboxymethyl cellulose edible coatings	a	16 days	Inhibition of fungal growth, improvement of shelf life	[28]
Minimally processed mangoes	Nisin-incorporated cellulose films	*L. monocytogenes*	12 days	Reduction *L. monocytogenes* by 1log	[158]
Pineapple	*Lactobacillus plantarum* UTNCys5-4 and *Lactococcus lactis* subsp. lactis Gt28	*E. coli*, *Salmonella* and *Shigella*	5 days	Reduction by 2.08, 1.43, and 1.91 log CFU/g, for *E. coli*, *Salmonella,* and *Shigella* respectively	[170]
Fresh tomatoes	*Lactobacillus plantarum* UTNGt2 coatings	*E. coli* and *Salmonella*	17 days	Inhibition	[171]
Fresh strawberries	*Lactobacillus plantarum* incorporated into carboxymethyl cellulose edible coatings	Yeast and molds	15 days	± color, hardness, TSS, TA, and total anthocyanin—weight loss, decay Less yeast and mold number	[172]
Fresh blueberries	*Lactobacillus rhamnosus CECT 8361* incorporated into alginate coatings	*Listeria innocua, E. coli O157:H7*	21 days	Reduction *L. innocua* counts by 1.7 log	[173]
Minimally processed papaya	Pediocin produced from *Pediococcus pentosaceus* incorporated alginate coating	Mesophilic bacteria and fungi	21 days	Inhibition of mesophilic bacteria and fungi	[174]

^a^ Not determined in the Publication.

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
