# Peer review of "Lactic Acid Bacteria as Antibacterial Agents to Extend the Shelf Life of Fresh and Minimally Processed Fruits and Vegetables: Quality and Safety Aspects"

_microorganisms, 2020, doi:10.3390/microorganisms8060952_

Round 1

Reviewer 1 Report

The manuscript microorganisms-834799 entitled “Lactic acid bacteria as antimicrobial agents to extend the shelf-life of fresh and minimally processed fruits and vegetables: quality and safety aspects” is clearly organized and the subject matter is suitable for publication in this journal. Some suggestions are listed below:

  • Line 18: Verify the word ‘bioreservation’
  • Lines 138-145: Bifidobacteria are not LAB, verify this along the manuscript and just cite LAB.
  • In figure 1 and 2 change the writing in white to black in order to improve visualization.
  • Lines 186-190: do not make paragraphs with just one sentence, check the textual cohesion.
  • Remove italics from Salmonella serotypes (Line 213, line 243, line 263, …)
  • Line 254: Verify citation style
  • Line 436: the authors are implying that bifidobacteria are LAB, is better to clarify or remove.
  • Nothing was discussed about molds and yeasts even knowing that these microorganisms are very important as contaminants and deteriorators in fruits and vegetables. I suggest adding discussion about LAB against fungi or changing the title and focus to antibacterial and not to antimicrobial.

Other than that the manuscript is well discussed and could be approved for publication.

Author Response

First of all thank you very much for the careful analysis of the manuscript and sending a detailed review. Detailed answers to your comments are given below. We tried to improve the article according to your detailed comments. We highlighted all changes in the manuscript using the "Track Changes" function in Microsoft Word. 

Line

Comment

Answer

Line 18:

·        Verify the word ‘bioreservation’

It was changed into: "bioprotection".

Lines 138-145:

·         Bifidobacteria are not LAB, verify this along the manuscript and just cite LAB.

Bifidobacteria has been removed.

·         In figure 1 and 2 change the writing in white to black in order to improve visualization.

It was corrected.

Lines 186-190:

·         do not make paragraphs with just one sentence, check the textual cohesion.

It was corrected.

·         Remove italics from Salmonella serotypes (Line 213, line 243, line 263, …)

It was corrected.

Line 254:

·         Verify citation style

It was corrected.

Line 436:

·         the authors are implying that bifidobacteria are LAB, is better to clarify or remove.

Bifidobacteria has been removed.

·         Nothing was discussed about molds and yeasts even knowing that these microorganisms are very important as contaminants and deteriorators in fruits and vegetables. I suggest adding discussion about LAB against fungi or changing the title and focus to antibacterial and not to antimicrobial.

Yes, we focused mainly on LAB which are mainly associated with foodborne outbreaks. So, we added the word antibacterial in the title.

Reviewer 2 Report

The paper entitled: Lactic acid bacteria as antimicrobial agents to extend the shelf-life of fresh and minimally processed fruits and vegetables: quality and safety aspects is well written and in my opinion important, however, some improvements are necessary. 

    1. Abstract: please checked the spelling "bioprotection" line 18
    2. re-phase the sentence lines 18-20. It is confusing.
    3. Introduction: re-phase the sentence line 88-91. Ozone, organic acids... are not alternatives to chlorine.
    4. lines 141-145. The genus Lactobacillus has been reclassified, according to a new publication, the authors should check and correct the bacterial names in the text accordingly. A web-based tool has been developed to help to determine the new names of all Lactobacillus species http://lactotax.embl.de/wuyts/lactotax/
    5. line 172: checked spelling ".... such as..." not "such us"
    6. lines 196-198 and 220-221. Please add a comparison to European Union cases. I believe it could be interesting to the readers.
    7. line 303-304 and 324 ".... mainly against Gram-positive bacteria". Please avoid repeating information.
    8. I believe at Conclusion section you should more focus on the discussion on the pros and cons of  bacteriocin and LAB films possibility of use in fresh-cut fruits and vegetables industry 

Author Response

First of all thank you very much for the careful analysis of the manuscript and sending a detailed review. Detailed answers to your comments are given below. We tried to improve the article according to your detailed comments. We highlighted all changes in the manuscript using the "Track Changes" function in Microsoft Word.

Line

Comment

Answer

18

Abstract: please checked the spelling "bioprotection" line 18

It was changed into: "bioprotection".

lines 18-20.

re-phase the sentence lines 18-20. It is confusing.

This sentence was reworded: The use of lactic acid bacteria in raw and minimally processed fruits and vegetables helps to better maintain their quality by extending their shelf life, causing a significant reduction and inhibition of the action of important foodborne pathogens.

line 88-91

Introduction: re-phase the sentence line 88-91. Ozone, organic acids... are not alternatives to chlorine.

This sentence was reworded: Different chemical alternatives to chlorine have been used, such as, chlorine dioxide, and acidified sodium chlorite. Moreover other substances have been used for the same purpose, such as, ozone, organic acids, peroxyacetic acid, hydrogen peroxide, electrolysed water, and calcium-based solutions.

lines 141-145.

The genus Lactobacillus has been reclassified, according to a new publication, the authors should check and correct the bacterial names in the text accordingly. A web-based tool has been developed to help to determine the new names of all Lactobacillus species http://lactotax.embl.de/wuyts/lactotax/

The new publication was added.

line 172:

checked spelling ".... such as..." not "such us"

It was changed into: such as.

lines 196-198 and 220-221.

Please add a comparison to European Union cases. I believe it could be interesting to the readers.

Data from European Union was added.

line 303-304 and 324 "....

mainly against Gram-positive bacteria". Please avoid repeating information.

The sentence in line 324 was removed.

I believe at Conclusion section you should more focus on the discussion on the pros and cons of  bacteriocin and LAB films possibility of use in fresh-cut fruits and vegetables industry 

Pros and cons of bacteriocin and LAB films were added.